# Score-Based Black-Box Adversarial Attack on Time Series Using Simulated Annealing Classification and Post-Processing Based Defense

Sichen Liu [1,2] and Yuan Luo [1,2,*]

1 Department of Computer Science and Engineering, Shanghai Jiao Tong University, Shanghai 200240, China; dyielsc@sjtu.edu.cn
2 Blockchain Advanced Research Center, Shanghai Jiao Tong University, Wuxi 214104, China
* Correspondence: yuanluo@sjtu.edu.cn

**Abstract:** While deep neural networks (DNNs) have been widely and successfully used for time series classification (TSC) over the past decade, their vulnerability to adversarial attacks has received little attention. Most existing attack methods focus on white-box setups, which are unrealistic as attackers typically only have access to the model's probability outputs. Defensive methods also have limitations, relying primarily on adversarial retraining which degrades classification accuracy and requires excessive training time. On top of that, we propose two new approaches in this paper: (1) A simulated annealing-based random search attack that finds adversarial examples without gradient estimation, searching only on the $l_\infty$-norm hypersphere of allowable perturbations. (2) A post-processing defense technique that periodically reverses the trend of corresponding loss values while maintaining the overall trend, using only the classifier's confidence scores as input. Experiments applying these methods to InceptionNet models trained on the UCR dataset benchmarks demonstrate the effectiveness of the attack, achieving up to 100% success rates. The defense method provided protection against up to 91.24% of attacks while preserving prediction quality. Overall, this work addresses important gaps in adversarial TSC by introducing novel black-box attack and lightweight defense techniques.

**Keywords:** time series classification; adversarial attack; adversarial attack defense

## 1. Introduction

Time Series Classification (TSC) has become a popular topic with the development of sensor technology, and can have benefits in scenarios in health care, power consumption monitoring, and to industrial observations [1,2]. In the past decade, several Deep Neural Network (DNN)-based methods such as InceptionTime [3], ResNet [1] and TapNet [4] have been proposed to solve the problem and achieve high performance. However, DNN is vulnerable to adversarial attacks, where small and imperceptible perturbations added to clean samples can mislead the classifier to give wrong predictions [5].

Adversarial attacks on DNNs are divided into white-box attacks and black-box attacks based on whether the attacker obtains model information. If all the information such as the model structure, training samples, model parameters and so on, of the victim model is revealed to the attacker, the attack approach is called a white-box attack, while a black-box attack can only access the predicted label or the confidence scores of all classes [6]. Besides, there are also grey-box attacks where only part of the information is available compared to white-box attacks. Gradient-based attacks like the Fast Gradient Sign Method (FGSM) [7], Basic Iterative Method (BIM) [8], Projected Gradient Descent (PGD) [9], C&W(Carlini and Wagner) [10] and so on calculate the gradient of loss and craft perturbation along the upward direction of loss function. Score-based attacks exploit the confidence score of every class. Attackers estimate the gradient through the confidence

scores [11–13] or randomly search perturbations to minimize the margin loss [14]. Besides, decision-based attacks can only get the predicted label. To defend adversarial attacks, most works train a robust model through adversarial training [9,15,16] or perform data pre-processing [17–20], and few works have focused on dynamic inference [21,22] and score post-processing [5].

Adversarial attacks and defense mainly focus on Image Classification and have been studied quite thoroughly in the field. Although more and more DNN models are applied to TSC, few researchers work on the adversarial attack of DNN models used for TSC, and even less works have focused on corresponding defense. The adversarial attack and corresponding defense are demonstrated in Figure 1. Papers such as [2,23] propose gradient-based methods in white-box settings which require the gradient of the model, and it is unrealistic because the classifier is usually a block-box to attackers. Therefore, we focus on the score-based attack in black-box setting and corresponding defense approach, where only predicted confidence scores are needed for both attacks and defense. Black-TreeS [13] adopts tree search strategy to find important positions and estimate gradients at the selected positions. TSadv [24] solves an optimization problem using the differential evolution algorithm without estimating the gradient. In [25], an adversarial transformation network on a distilled model is utilized to perform both black-box and white-box attacks. Existing defense against adversarial attacks on TSC adopts adversarial training strategy which adds adversarial examples to the training set and retrain the model [23,25,26]. However, adversarial training leads to high training cost and accuracy reduction by enlarging the original training set.

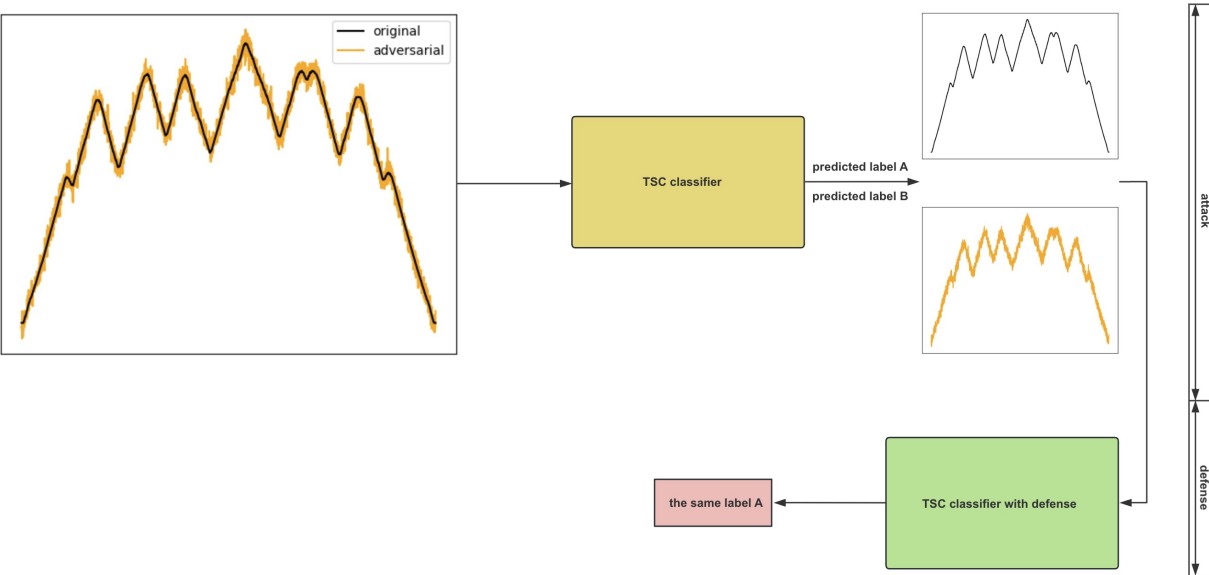

**Figure 1.** Adversarial attack and corresponding defense on TSC.

In view of these limitations, we propose square-based black-box adversarial attack and defense approaches on TSC. In terms of attack, we focus on the black-box setting where attackers can only access the confidence scores. Instead of estimating the gradient of the classifier, a simulated annealing-based random search method is adopted to find the adversarial examples minimizing the margin loss. In terms of defense, a post-processing-based defense strategy is proposed. The post-processing module takes the output confidence scores as inputs and flips the trend of the margin loss periodically. However, the global trend remains unchanged. Overall, the contributions of this paper are mainly as follows:

- We propose a more realistic scored-based black-box attack approach on TSC through simulated annealing-based random search algorithm without gradient estimation.

- We propose a post-processing-based defense approach against scored-based black-box attacks on TSC where only output confidence scores are needed. The trend of the loss function is flipped locally while the global trend of the loss function is not changed. Besides, the accuracy of the classifier is not affected.
- We carry out experiments on multiple time series datasets to demonstrate the effectiveness of both the attack and defense approaches.

## 2. Materials and Methods

### 2.1. Background

In this paper, only a univariate time series was considered. A time series is a sequence of data indexed in time order, and we can describe a time series as a vector $X = [x_1, x_2, \ldots, x_t] \in \mathbb{R}^t$. Each time series has a label $K$, which indicates the class of the time series. The goal of TSC is to get a classifier $f : \mathbb{R}^t \to \mathbb{R}^{\mathcal{K}}$ mapping input time series $x$ of length $t$ to probabilities of $x$ belonging to the $\mathcal{K}$ classes.

Given the confidence scores $Y = f(X) \in \mathbb{R}^{\mathcal{K}}$, the predicted class $K$ of the classifier is $K = \underset{K'=1,2,\ldots,\mathcal{K}}{\arg\max} f_{K'}(X)$. The target of score-based black-box adversarial attack is to find a sample $X'$ within the $l_\infty$-norm ball of radius $\epsilon$ that satisfies:

$$\underset{K'=1,2,\ldots,\mathcal{K}}{\arg\max} \ f_{K'}(X') \neq K, \qquad \|X' - X\|_\infty \leq \epsilon. \tag{1}$$

If the predicted class $K'$ is determined, it is called a targeted attack, while in an untargeted attack, the value of $K'$ is arbitrary except for $K$. In this paper, we only focus on untargeted attack under $l_\infty$-norm restriction since the computational complexity is quite low. The target can be transformed to an optimization problem minimizing the margin loss $L_{margin}$:

$$\min_{X'} \ L_{margin}(f(X'), K) = \min_{X'} \ f_K(X') - \max_{K' \neq K} f_{K'}(X'), \qquad \|X' - X\|_\infty \leq \epsilon, \tag{2}$$

where $f_{K'}(X')$ represents the confidence scores of sample $X'$ belonging to class $K'$ given by classifier. The optimization step terminates when $L_{margin}$ drops below 0. The target of the defense is to prevent attackers from finding qualified samples.

Simulated annealing is a probabilistic technique for approximating the global optimum of a given function. Given the objective function $g(x)$ and the starting point $x_0$, we randomly select a neighbor $x_1$ of $x_0$ each time. Then we calculate the values of both $g(x_0)$ and $g(x_1)$ and the amount of change $\Delta g = g(x_1) - g(x_0)$. For minimization problems, we update $x_0$ to $x_1$ with a probability $\min(e^{-\Delta g/T}, 1)$, where $T$ is a hyperparameter called temperature which declines with the iterations.

### 2.2. Methods

#### 2.2.1. Square-Based Attack

Our attack is generally based on the random search algorithm which is a family of numerical optimization methods that do not require the gradient of the problem. This differentiates it from classical black-box attack methods which typically estimate the gradient of the problem. We adopt a simulated annealing algorithm which belongs to the random search algorithm to reach the goal in (2). Compared to the simplest random search algorithm of hill climbing, simulated annealing is more able to jump out of the local optimum. The main idea of our method is to sample a random noise vector $\delta$ within the $l_\infty$-norm ball of radius $\epsilon$ and add it to current $X'$ to form $X''$. If $X''$ improves the objective function, we update $X'$ to $X''$, and on the contrary, we update $X'$ to $X''$ with certain probability.

Unlike classical random search methods that search in the $l_\infty$-norm ball of radius $\epsilon$ for candidate vector $\delta$, we just search on the boundary of the $l_\infty$-norm ball. In other words, the search space satisfies $\|\delta\|_\infty = \epsilon$. Besides, we just modify a decreasing fraction $p$ of the

perturbation vector $\delta$ continuously. In this way, the changes of every step are localized and maximized under the assumption that successful $l_\infty$-perturbations usually have values $\pm\epsilon$ in all the components [14].

The scheme of our algorithm is demonstrated in Algorithm 1. Before the iterations begin, we initialize the perturbation vector $\delta$ by sampling uniformly from $\pm\epsilon$. First, the algorithm selects the length $w$ of the continuously changing part of $\delta$ based on a piecewise decreasing hyperparameter $p$, and sample uniformly from $\pm\epsilon$ for $w$ times. Then we replace an arbitrary continuous sequence of length $w$ in vector $\delta$ with the sampled values, and a new sample $X''$ is derived from adding $\delta$ to $X$. We accept the new sample with a probability if the value of objective function increases, and if the value drops, the new sample is accepted. When the new sample is accepted, we update $X'$ to $X''$. Finally, we update the value of two hyperparameters. More specifically, the value of $p$ reduces by half at iteration $n \in \{10, 50, 100, 200, 500, 1000, 2000, 4000, 8000\}$, and the value of $T$ changes every iteration with the attenuation rate $\gamma$ to $\gamma T$.

---

**Algorithm 1:** Simulated annealing-based adversarial attack

>**Input** : classifier $f$, time series $X$, label $K$, radius of $l_\infty$-norm ball $\epsilon$, length of time series $s$, fraction of change $p(0 < p < 1)$, fraction decline function $d_p$, temperature $T$, temperature decline function $d_T$, temperature attenuation rate $\gamma(0 < \gamma < 1)$ and number of iterations $N$
>
>**Output**: adversarial example $X'$

1   $\delta \leftarrow \text{Uniform}(\{-\epsilon, \epsilon\})^l, \;\; X' \leftarrow X + \delta, \;\; i \leftarrow 1$;
2   **while** $i \leq N$ **and** $X'$ *is not adversarial* **do**
     /\* replace any continuous sequence of length $w$ in vector $\delta$ with uniformly sampled values                    \*/
3     $w \leftarrow \lfloor p*l \rfloor, \;\; \delta_p \leftarrow \text{Uniform}(\{-\epsilon, \epsilon\})^w$ ;
     /\* $s$ is the starting index of the continuous sequence replaced    \*/
4     $s \leftarrow \text{Uniform}(\{0, 1, \ldots, l - w\})$ ;
5     $\delta_{s:s+w} \leftarrow \delta_p$ ;
6     $X'' \leftarrow X + \delta$ ;
     /\* simulated annealing part                                           \*/
7     $\Delta L \leftarrow L_{margin}(f(X''), K) - L_{margin}(f(X'), K)$ ;
8     **if** $\Delta L < 0$ **or** $\text{Uniform}([0,1]) < e^{\frac{-\Delta L}{T}}$ **then**
9       $\mid \;\; X' \leftarrow X''$
10    **end**
11    $i \leftarrow i + 1, \;\; p \leftarrow d_p(p), \;\; T \leftarrow d_T(T) = \gamma T$ ;
12 **end**

---

### 2.2.2. Post-Processing-Based Defense

Traditional defense methods like adversarial training retrain the model with adversarial examples, which affects the accuracy of the classifier and suffers from high training cost. Our defense strategy encapsulates the defense function into a post-processing module independent from the classifier. It means we only slightly adjust the output confidence scores while the accuracy of the classifier is not affected. Besides, since the post-processing module is independent from the classifier, the training steps do not involve parameters of the classifier. Therefore, the training cost is low.

Black-box attacks iterate along the decreasing direction of the objective function no matter whether the attack is based on gradient estimation or random search. Therefore, the main idea of our approach is to modify the confidence scores which flips the trend of the objective function. Along the direction of adversarial attack, the value of the objective function increases. As a result, the attacker cannot achieve the adversarial examples. However, our aim is to mislead the attackers instead of the users. In order to minimize the difference of confidence scores, the global trend of the objective function

should be preserved. Therefore, we flip the trend piecewise and periodically to get adversarial examples.

$L_{margin}$ in (2) is adopted as the objective function in this paper, and for each confidence score vector and label predicted by the classifier $f$, the value of loss is $L_{margin}(f(X), K)$, denoted as $l_s$. Based on Chain rule, the trend of

$$l_d = \beta - \alpha * l_s, \qquad \beta \in [0,1], \alpha > 0 \tag{3}$$

is opposite to the trend of $l_s$. Let $P$ be the predicted confidence score and $l_s = g(P)$. The derivative of $l_s$ can be expressed as $g'(P)$, and the derivative of $l_d$ is $-\alpha * g'(P)$ based on Chain rule. Under the condition that $\alpha > 0$, the derivative of $l_d$ has the opposite sign of the derivative of $l_s$. Therefore, when proper values are assigned to $\alpha$ and $\beta$, we can obtain the constructed confidence scores of $l_d$ through gradient descent. We divide the value of $l_s$ ranging from 0 to 1 into intervals of length $t$, and the interval ranges from $\lfloor l_s/t \rfloor * t$ to $(\lfloor l_s/t \rfloor + 1) * t$. Let the midpoint of the interval be $l_m$, and it can be expressed as

$$l_m = (\lfloor l_s/t \rfloor + 1/2) * t. \tag{4}$$

The value of $l_m - l_s$ decreases from $t/2$ to $-t/2$ as $l_s$ and increases from $n * t$ to $(n + 1) * t$, but the value ranges are the same across different intervals. It does not meet the requirement that the global trend of the objective function is preserved. The trend of $l_m - l_s$ is opposite to that of $l_s$ and adding a constant term to $l_m - l_s$ does not change the property within the interval. To preserve the global trend, the constant term should increase as the value of $l_s$ increase to another interval, and $l_m$ satisfies the requirement. Thus, we can add $l_m$ to $\alpha * (l_m - l_s)$, constructing

$$l_d = l_m - \alpha * (l_s - l_m), \tag{5}$$

and it is in accordance with (3) with $\beta = (1 + \alpha) * l_m$. If $\alpha = 1$, $l_d$ and $l_s$ have the same domain, and the value of $l_d$ increases as $l_s$ increases to a value in the next interval.

To mislead attackers, the margin loss $L_{margin}$ calculated from the adjusted confidence scores should be close to $l_d$. Besides, to reserve the prediction confidence, the change of the largest probability should be minimized. Thus, the target is to solve the optimization problem:

$$\min_{P'} \; |L_{margin}(\text{softmax}(P'), K) - l_d| + \mu \cdot |\text{softmax}_K(P') - P_K|, \tag{6}$$

where $P$ and $P'$ are the original and perturbed confidence scores and $\mu$ is the hyperparameter that balance between the two objectives.

The scheme of our algorithm is demonstrated in Algorithm 2. First, we calculate the margin loss of the predicted confidence scores $l_s$, and we can calculate the constant term $l_m$ along with the constructed loss $l_d$ based on $l_s$. We then optimize the modified confidence score vector with two objectives: (1) minimizing the difference between the margin loss of the constructed confidence score vector and the constructed loss, and (2) minimizing the change of the highest probability value. The algorithm is mainly designed to defend the attack method proposed in previous sections. Since the attack method is also a black-box to the defense method where the defense method is agnostic to attack details, it can also defend other score-based black-box attacks. Besides, a confidence score vector $P'$ whose margin loss is close to the value of $l_d$ is generated given the original margin loss $l_s$. Moreover, the highest probability value of $P'$ is close to that of the original confidence score vector $P$.

It is easy to prove the effectiveness of our approach. Let $A$ denote a point on the curve of $l_d$ in Figure 2. The attacker finds another point $B$ in the neighbourhood of $A$, so $B$ either locates in the same interval of $A$ or the adjacent intervals. If $B$ locates in the same interval of $A$, the attacker accepts $B$ when $B$ is on $A$'s left side (constructed margin loss decreases). If B is located in another interval, the attacker accepts B if B falls within the interval to the right

of A's interval. Thus, the attack either succeeds when jumping out of the rightmost interval or fails when converging to the leftmost point of an interval. Actually, it is difficult to jump into the right interval, let alone the rightmost one, especially when the perturbation is small and the period is large.

---

**Algorithm 2:** Post-processing-based adversarial attack defense

    **Input**    : confidence scores predicted by classifier $P$, label $K$, number of iterations
             $N$ and hyperparameters period $t$, $\alpha(\alpha > 0)$ and $\mu(\mu > 0)$
    **Output**: modified confidence score vector $P'$

1  $l_s \leftarrow L_{margin}(P, K)$;
    /* $l_m$ is the midpoint of the interval                            */
2  $l_m = (\lfloor l_s/t \rfloor + 1/2) * t$;
    /* $l_d$ is the constructed loss                                  */
3  $l_d = l_m - \alpha * (l_s - l_m)$;
4  $P' \leftarrow P$;
5  optimize $P'$ with loss function
    $|L_{margin}(\text{softmax}(P'), K) - l_d| + \mu \cdot |\text{softmax}_K(P') - P_K|$ for $N$ epochs

---

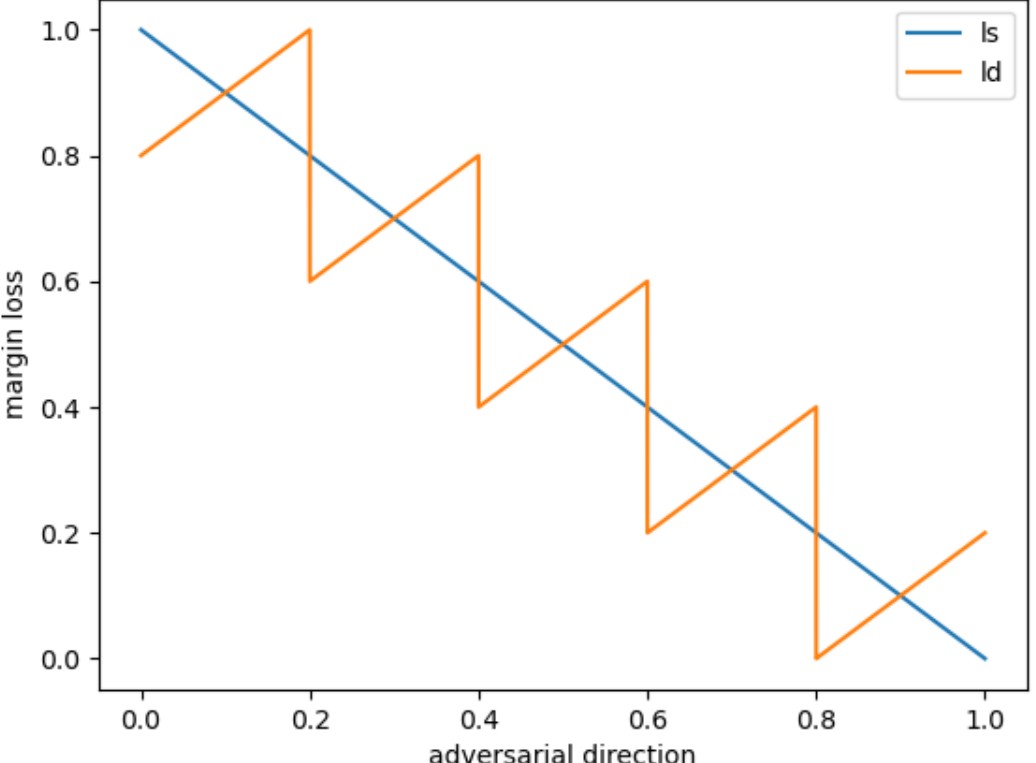

**Figure 2.** The original margin loss $l_s$ and constructed margin loss $l_d$ when $t = 0.2$, $\alpha = 1$. The trends of $l_s$ and $l_d$ are the same globally, but they are opposite locally.

### 2.3. Experiments

We select InceptionTime as the TSC classifier, and conduct experiments on four UCR datasets: UWaveGestureLibraryAll, OSU Leaf, ECG5000 and ChlorineConcentration. We evaluate the effectiveness of our attack and defense methods with difference settings of the $\epsilon$ value. The evaluation matrices are

$$ASR = NAS/NTS,$$
$$DSR = NDS/NAS,$$
$$AQT = \sum_{x \in AS} QT_x/NAS, \tag{7}$$
$$MQT = median(QT),$$

where $ASR$ is the average success rate, $NAS$ is the number of adversarial examples, $NTS$ is the number of all samples, $NDS$ is the number of defended adversarial examples, $AS$ is the adversarial example set, $QT_x$ is the query times of sample $x$ until the attack succeeds, $AQT$ is the number of average query times of successful adversarial attacks and $MQT$ is the number of median query times of successful adversarial attacks.

The parameter setting of InceptionTime is kernel size 40, number of filters 32, bottleneck size 32 and depth 6. We evaluate the performance of attack and defense approaches with $\epsilon$ ranging in $\{0.05, 0.1, 0.15\}$ to observe the impact of perturbation intensity. The initial value of the fraction of change $p$ is 0.05, and the initial value of the temperature $T$ is 100 with decay rate 0.99. In terms of defense, we set the period $t$ to 0.05, hyperparameter $\alpha$ to 1 and hyperparameter $\mu$ to 1, and the objective function is optimized for 250 epochs. All the experiments are conducted on a machine with 9 NVIDIA Tesla V100 GPUs with 32 GBs memory(only one GPU is needed for each experiment, and the GPU utilization approaches 100%). We exploit the sktime package to build the DNN model and implement the attack and defense approaches through TensorFlow. The version of core software is python 3.10.4, sktime 0.24.1, tensorflow 2.14.0 and numpy 1.26.2.

## 3. Results

Table 1 shows the classification accuracy of the TSC classifier and the number of classes of the four datasets: UWaveGestureLibraryAll, OSU Leaf, ECG5000 and ChlorineConcentration. The accuracy of the TSC classifier InceptionTime exceeds 85% for all four datasets with more than two classes.

**Table 1.** TSC classifier accuracy and number of classes of different datasets.

| Dataset | TSC Classifier Accuracy | Number of Classes |
|---|---|---|
| UWaveGestureLibraryAll | 95.20% | 8 |
| OSU Leaf | 94.21% | 6 |
| ECG5000 | 94.09% | 5 |
| ChlorineConcentration | 87.66% | 3 |

Figure 3 shows the ASR with $\epsilon$ ranging in $\{0.05, 0.1, 0.15\}$ over successive iterations, and (a–d) correspond to the results of four UCR datasets. It is clear that as the $\epsilon$ increases, the ASR also increases accordingly. The ASR from the highest to the lowest is: ChlorineConcentration, UWaveGestureLibraryAll, OSU Leaf and ECG5000.

Figure 4 shows the AQT with $\epsilon$ ranging in $\{0.05, 0.1, 0.15\}$ over successive iterations, and (a-d) correspond to the results of four UCR datasets.

Figure 5 shows the DSR with $\epsilon$ ranging in $\{0.05, 0.1, 0.15\}$ of the four datasets: UWaveGestureLibraryAll, OSU Leaf, ECG5000 and ChlorineConcentration. It is clear that as the value of $\epsilon$ increases, the DSR decreases accordingly.

Table 2 shows the performance of our proposed attack and defense methods across four different datasets under varying values of $\epsilon$, and the table reports the results on four main evaluation metrics: ASR, AQT, MQT and DSR.

We compare the ASR of our approach with existing adversarial attack methods on UWave dataset setting $\epsilon$ to 0.3, and the results are shown in Table 3.

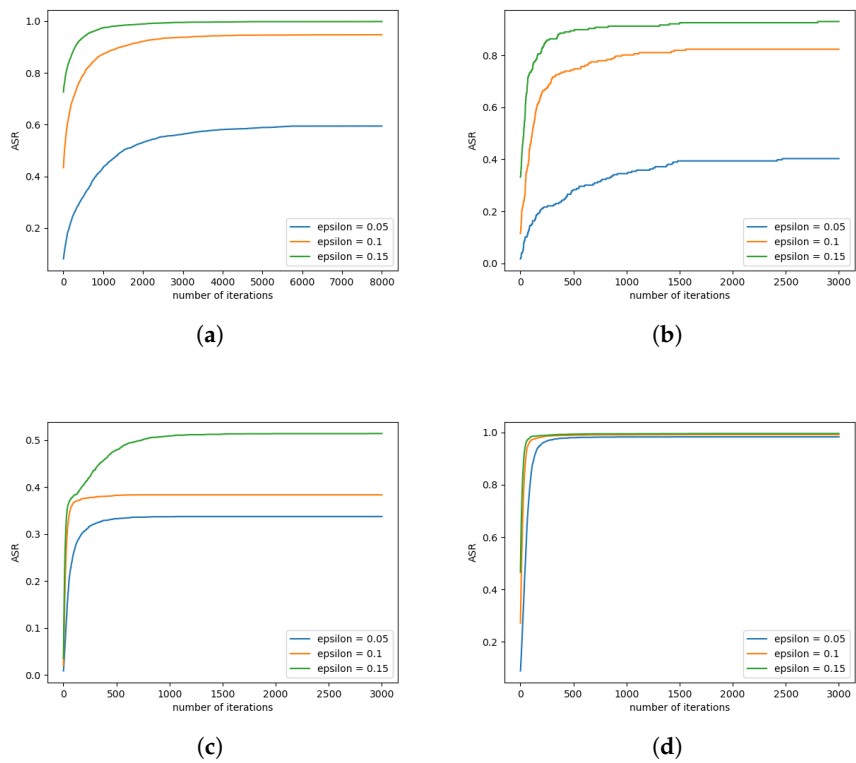

**Figure 3.** ASR curve of different $\epsilon$ values. Each figure shows the result of one dataset. (**a**) UWaveGestureLibraryAll; (**b**) OSU Leaf; (**c**) ECG5000; (**d**) ChlorineConcentration.

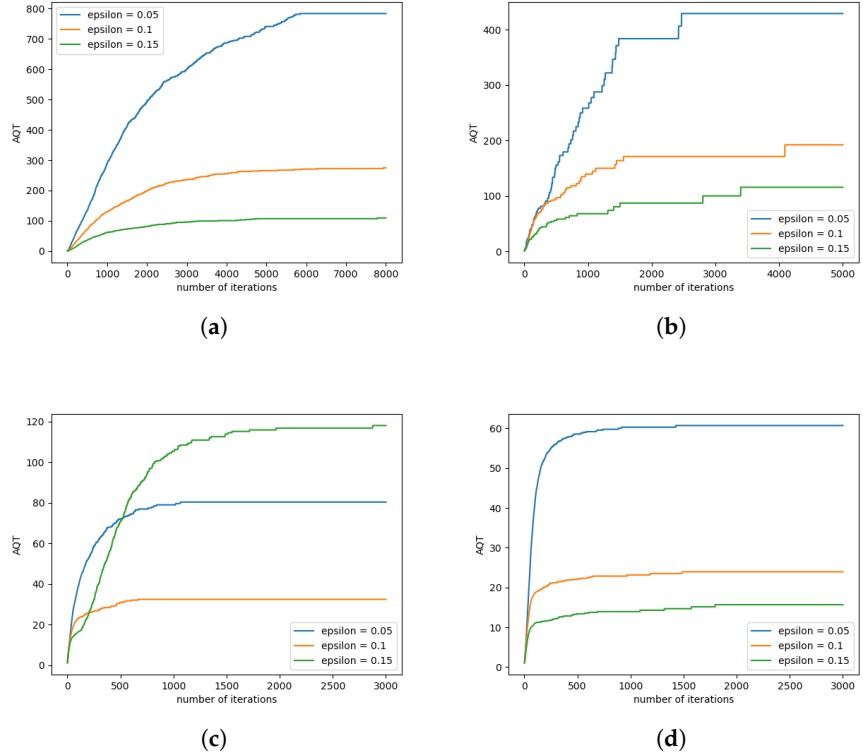

**Figure 4.** AQT curve of different $\epsilon$ values. Each figure shows the result of one dataset. (**a**) UWaveGestureLibraryAll; (**b**) OSU Leaf; (**c**) ECG5000; (**d**) ChlorineConcentration.

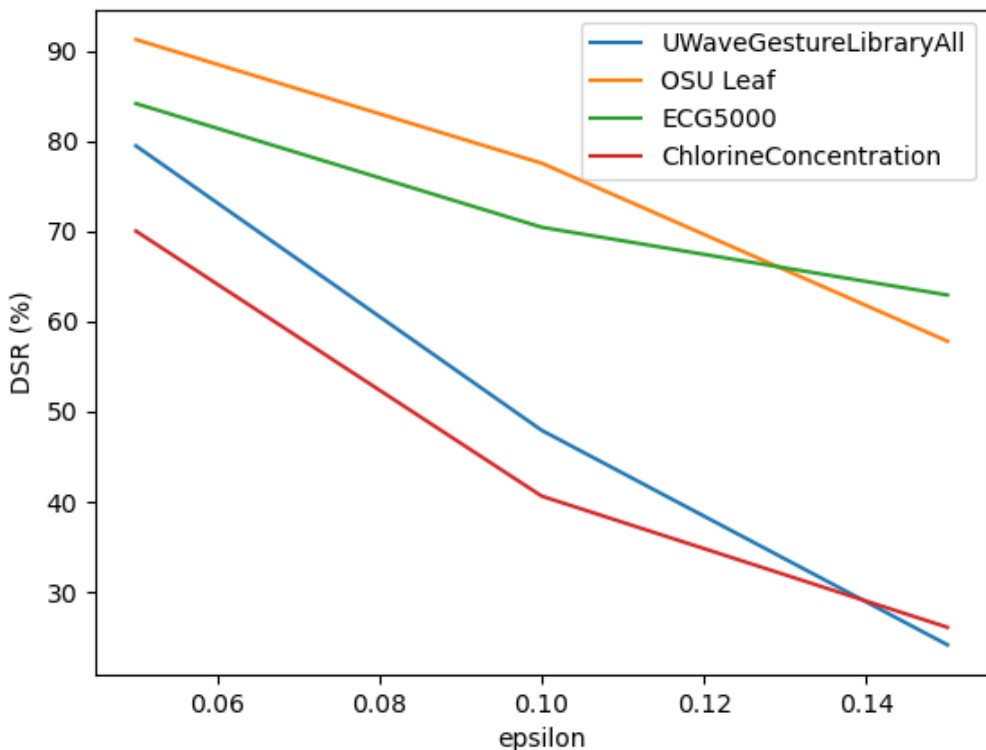

**Figure 5.** DSR curve of different $\epsilon$ values in four datasets: UWaveGestureLibraryAll, OSU Leaf, ECG5000 and ChlorineConcentration.

**Table 2.** ASR, AQT and DSR of different datasets with different $\epsilon$ values.

| Dataset | $\epsilon$ Value | ASR | AQT | MQT | DSR |
|---|---|---|---|---|---|
| UWaveGestureLibraryAll | 0.05 | 59.45% | 781.03 | 403 | 79.46% |
| | 0.1 | 94.96% | 277.48 | 17 | 47.94% |
| | 0.15 | 99.91% | 112.20 | 1 | 24.17% |
| OSU Leaf | 0.05 | 40.27% | 429.33 | 187 | 91.24% |
| | 0.1 | 82.74% | 192.35 | 84 | 77.54% |
| | 0.15 | 93.36% | 115.80 | 26 | 57.82% |
| ECG5000 | 0.05 | 33.75% | 80.34 | 45 | 84.15% |
| | 0.1 | 38.36% | 32.45 | 19 | 70.44% |
| | 0.15 | 51.41% | 118.01 | 17 | 62.93% |
| ChlorineConcentration | 0.05 | 98.28% | 60.70 | 48 | 70.02% |
| | 0.1 | 99.13% | 23.94 | 13 | 40.65% |
| | 0.15 | 99.55% | 15.68 | 4 | 26.11% |

**Table 3.** Comparison of ASR with existing adversarial attack methods on UWave dataset.

| Matrix \ Dataset | FGSM | PGD | NES [11] | BlackTreeS [13] | Our Approach |
|---|---|---|---|---|---|
| ASR | 43.2% | 58.0% | 17.1% | 100% | **100% *** |

* Our attack achieves the ADR of 100% when $\epsilon = 0.2$, and the ADR drops to 97.57% when $\epsilon = 0.3$.

## 4. Discussion

If an attack method is able to achieve the same ASR against a more robust model, then this indicates a higher level of effectiveness of the attack method. If a classifier achieves high accuracy on its training and validation data, this can be an indication that it is more robust when facing adversarial attacks. The reason is that a model that learns the underlying patterns in the data very well, resulting in strong predictive performance,

has gained a deeper understanding of the legitimate sample space. This makes it less susceptible to being fooled by small perturbed variations that move examples outside that natural data distribution. Therefore, we select datasets where the classifier achieves high validation accuracy after training. The robustness of a classifier depends not only on the accuracy achieved through training, as aforementioned, but also on the number of training samples used. Insufficient samples can hinder a model's ability to fully learn intricacies of different classes. On top of that, we select the aforementioned four datasets from UCR archive.

In terms of model selection, a classifier with higher accuracy indicates that an attack would be more effective at reaching the same ASR. Therefore, we choose the state-of-the-art InceptionNet model which achieves high classification accuracy as the classifier. Evaluation of our approaches using other classifiers can be left for future research. We choose the margin loss as the loss function because it directly optimizes the objective, and papers on score-based black-box attacks in image classification domain also adopt this loss function. We set the hyperparameter $\lambda$ to 1, which means the two objectives are given equal importance. Additionally, the first term represents the difference in margin loss and the second term represents the difference in probability. Both terms have the same value domain, so they have similar impact on the objective function. A detailed comparison of different $\lambda$ values can be left for future research. The number of optimization epochs is determined during the optimization process, and it should be dynamically adjusted for different datasets according to how quickly the optimization objective converges. However, a large number of epochs does not lead to overfitting, since this is only an optimization problem rather than one of model fitting.

From Figure 3, we can see that as the value of $\epsilon$ increases, representing a larger allowable perturbation, the ASR also increases as expected. Intuitively, a larger value of $\epsilon$ means it is easier for the adversarial example to cross decision boundaries and fool the classifier. Besides, larger $\epsilon$ values imply a higher starting speed due to a more relaxed constraint, but it does not necessarily translate to faster convergence. That's because a larger value of $\epsilon$ corresponds to a larger search space within which the adversarial perturbation can vary. As the footnote to Table 3 indicates, even though a larger $\epsilon$ value of 0.3 allows for greater flexibility, our attack method achieves better results with the $\epsilon$ value of 0.2 on that dataset. This suggests that while a bigger $\epsilon$ value opens up a wider search space, it also increases the chance of getting stuck in local minima when optimizing for the attack objective.

Existing black-box attacks on TSC have not focused on the query times of the classifier. However, query efficiency is an important metric to evaluate attack performance, as it measures how fast adversarial examples can be generated. Figure 4 shows AQT needed to find successful adversarial examples for different $\epsilon$ values. We can see that generally, a larger $\epsilon$ value requires fewer queries as the larger value of $\epsilon$ means attacks are easier to succeed. However, Figure 4c is an exception because there is a special sample, but the finding can be justified if we observe the MQT. As the value of $\epsilon$ increases, allowing larger perturbations, the AQT converges more slowly due to the expanded change of jumping out of local minima. Besides, the increasing rate of AQT decreases with larger $\epsilon$ values, because fewer adversarial examples can be discovered in the same time period given the larger search space, which decreases the dividend when calculating the AQT.

Table 3 shows that the proposed approach achieves a higher ASR than most existing time series attack methods, demonstrating better attack performance. Critically, it reaches the optimal ASR while using a smaller $\epsilon$ value than comparative methods. In addition, the number of queries needed is comparatively low for the proposed approach. Together, achieving top ASR with less perturbation severity (smaller $\epsilon$ values) and faster optimization (lower queries) suggests the proposed approach introduces a better balance of attack strength and query efficiency. Therefore, the low query counts demonstrated by the proposed method can be used as a new performance baseline when evaluating time series attack algorithms.

Figure 5 shows that the DSR decreases as the value of $\epsilon$ increases because larger $\epsilon$ values increases the defense difficulty. For the UWaveGestureLibraryAll and ChlorineConcentration datasets with the $\epsilon$ value of 0.15, the ASR is almost 100% while the DSR remains very low. This suggests that when some perturbation vectors are initialized under this large $\epsilon$ value, they immediately form adversarial examples. The MQT is also very small in this setting, with over half of adversarial samples constructed within 4 queries or less. This low MQT provides further evidence that the large $\epsilon$ value of 0.15 is sufficient for the perturbation to exceed the decision boundary in the first attempt for many samples, resulting in the very high ASR and confirming our hypothesis.

Compared to existing TSC defense methods that use adversarial training, our defense method has two advantages. First, adversarial training adds adversarial examples to the training set, which decreases accuracy of clean samples, especially as the number of samples increases. Second, enlarging the training set increases training costs such as time and resource consumption. However, adversarial training enhances the classifier, while our defense method leaves the classifier unchanged. Thus, adversarial training can distinguish "immediate" adversarial samples generated right after attack begins.

### 5. Future Research

Some potential directions for future research include:

- Exploring additional random search methods like genetic algorithms as alternatives to simulated annealing for perturbation optimization, and omparing the attack success rates of different methods and analyze their relative advantages and disadvantages.
- The current defense approach has a limitation in that it is not effective against adversarial examples where the starting perturbation already exceeds the range of the constructed loss curve. A potential area of improvement is developing a supplementary post-processing module capable of handling such "immediate" adversarial inputs, which is challenging given only access to confidence scores.

Overall, systematically comparing various random search attacks and exploring complementary defense mechanisms are able to handle extreme initial perturbations and could help strengthen adversarial machine learning techniques for both attack and defense.

### 6. Conclusions

In this paper, we propose a square-based black-box adversarial attack method along with a corresponding post-processing-based defense approach. We perform an attack using simulated annealing-based random search algorithm to find the adversarial examples on the $l_\infty$-norm hypersphere. The defense is based on a post-processing module only requiring the confidence scores predicted by a classifier, and the module flips the loss trend locally without affecting the global trend. We conduct experiments on four UCR datasets training with InceptionNet, and both attack and defense approaches achieve high performance. Performance of the proposed attack is also compared to existing adversarial attacks. Results show it achieves the best ASR while requiring few queries, indicating better efficiency. This optimal combination of high ASR and low query count can potentially be used as a new benchmark for evaluation of future time series adversarial attack methods. Additionally, this defense approach can be applied to strengthen TSC models designed for real-world scenarios such as ECG-based disease detection and industrial security monitoring.

**Author Contributions:** Conceptualization, S.L.; methodology, S.L.; software, S.L.; validation, S.L.; formal analysis, S.L.; investigation, S.L.; resources, S.L.; data curation, S.L.; writing—original draft preparation, S.L.; writing—review and editing, Y.L.; visualization, S.L.; supervision, Y.L.; project administration, Y.L.; funding acquisition, Y.L. All authors have read and agreed to the published version of the manuscript.

**Funding:** This research was funded by Shanghai Science and Technology Innovation Action Plan (No. 23511100400).



**Data Availability Statement:** We use the public dataset: UCR Time Series Classification Archive, and it can be accessed through the reference [27].

**Conflicts of Interest:** The authors declare no conflicts of interest.

**Abbreviations**

The following abbreviations are used in this manuscript:

TSC     Time Series Classification
DNN     Deep Neural Network
ASR     average success rate
DSR     defense success rate
AQT     average query times of successful adversarial attacks
MQT     median query times of successful adversarial attacks

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
