# Peer review of "Score-Based Black-Box Adversarial Attack on Time Series Using Simulated Annealing Classification and Post-Processing Based Defense"

_electronics, doi:10.3390/electronics13030650_

Round 1
Reviewer 1 Report
Comments and Suggestions for Authors
This paper focuses on the vulnerability of deep neural networks (DNNs) used for time series classification (TSC) to adversarial attacks. While DNNs have been successful in TSC, their susceptibility to attacks has received little attention. Existing attack methods often assume unrealistic access to the model's internal information, while defensive methods rely heavily on adversarial retraining, which can degrade accuracy and require extensive training time.
To address these limitations, the paper proposes two new approaches. Firstly, a simulated annealing based random search attack is introduced, which finds adversarial examples without gradient estimation. This attack operates within the l∞-norm hypersphere of allowable perturbations, making it more practical and effective. Secondly, a post-processing defense technique is presented, which periodically reverses the trend of loss values using only the classifier's confidence scores. This lightweight defense method maintains the overall trend while mitigating attacks. Experiments conducted on InceptionNet models trained on UCR dataset benchmarks demonstrate the effectiveness of the attack approach, achieving up to 100% success rates. The defense method provides protection against up to 91.24% of attacks while preserving prediction quality. Overall, this research addresses important gaps in adversarial TSC by introducing novel black-box attack and lightweight defense techniques, contributing to enhancing the security and robustness of DNNs in TSC tasks.
Based on a careful evaluation, it is clear that the paper requires some evisions to enhance its overall quality. Once these revisions are made, I believe that the paper will be suitable for publication.
1- The title should be revised to convey a more specific and concise meaning.
2- Authors should ensure proper punctuation throughout the manuscript.
3- Authors need to discuss and provide the sources of the data used in the paper, and consider including a source link in the footnote for reference.
4- A comparison with other attack methods should be included to provide a comprehensive analysis.
5- The results section should include more details to provide a thorough understanding of the findings.
6- More information about the software used, including the version or type of machine used for the analysis, should be provided.
7. It is recommended to collect the discussion and conclusions in the same section and should also suggest future research directions.
Comments on the Quality of English Language
This paper focuses on the vulnerability of deep neural networks (DNNs) used for time series classification (TSC) to adversarial attacks. While DNNs have been successful in TSC, their susceptibility to attacks has received little attention. Existing attack methods often assume unrealistic access to the model's internal information, while defensive methods rely heavily on adversarial retraining, which can degrade accuracy and require extensive training time.
To address these limitations, the paper proposes two new approaches. Firstly, a simulated annealing based random search attack is introduced, which finds adversarial examples without gradient estimation. This attack operates within the l∞-norm hypersphere of allowable perturbations, making it more practical and effective. Secondly, a post-processing defense technique is presented, which periodically reverses the trend of loss values using only the classifier's confidence scores. This lightweight defense method maintains the overall trend while mitigating attacks. Experiments conducted on InceptionNet models trained on UCR dataset benchmarks demonstrate the effectiveness of the attack approach, achieving up to 100% success rates. The defense method provides protection against up to 91.24% of attacks while preserving prediction quality. Overall, this research addresses important gaps in adversarial TSC by introducing novel black-box attack and lightweight defense techniques, contributing to enhancing the security and robustness of DNNs in TSC tasks.
Based on a careful evaluation, it is clear that the paper requires some evisions to enhance its overall quality. Once these revisions are made, I believe that the paper will be suitable for publication.
1- The title should be revised to convey a more specific and concise meaning.
2- Authors should ensure proper punctuation throughout the manuscript.
3- Authors need to discuss and provide the sources of the data used in the paper, and consider including a source link in the footnote for reference.
4- A comparison with other attack methods should be included to provide a comprehensive analysis.
5- The results section should include more details to provide a thorough understanding of the findings.
6- More information about the software used, including the version or type of machine used for the analysis, should be provided.
7. It is recommended to collect the discussion and conclusions in the same section and should also suggest future research directions.
Reviewer 2 Report
Comments and Suggestions for Authors
The developments presented in this interesting and well-documented submission is motivated by the fact that, during the past decade or so, the subject of deep neural networks (DNNs) have been widely and successfully used for the time-series classification (TSC). After convincingly remarking about some of the shortcomings in the methodology and techniques used in earlier works, the authors have successfully demonstrated some novel approaches for handling the involved problems. In my opinion, therefore, this submissions deserves to be published.
Reviewer 3 Report
Comments and Suggestions for Authors
This paper introduces a novel square-based black-box adversarial attack and a corresponding post-processing-based defense approach for time series classification (TSC) using deep neural networks (DNNs). The paper addresses the gaps in adversarial TSC by introducing a realistic black-box attack approach without gradient estimation and a lightweight defense strategy. Besides, the proposed attack achieves high success rates, and the defense method demonstrates robustness against adversarial attacks. However, there are a few suggestions for improvement:

Overall, the text is well-written and demonstrates high technical proficiency. However, there are a few suggestions for improvement:
1. Some sentences are quite complex and might benefit from being broken down into smaller, more digestible units. This can enhance overall readability.
2. The text appears to have formatting issues, especially in the algorithm sections. Consider revisiting the formatting to ensure proper indentation and clarity in the presentation of algorithms.
3. There are some minor grammatical issues and typos throughout the text. A careful proofread can help eliminate these, ensuring a polished final version.
Round 2
Reviewer 1 Report
Comments and Suggestions for Authors
After reviewing the revised version, I find it to be excellent, with all the comments thoroughly addressed. I am now confident in recommending it for publication.